# CYP7B1 as a Biomarker for Prostate Cancer Risk and Progression: Metabolic and Oncogenic Signatures (Diagnostic Immunohistochemistry Analysis by Tissue Microarray in Prostate Cancer Patients—Diamond Study)

**DOI:** 10.3390/ijms25094762

**Published:** 2024-04-27

**Authors:** Giorgio Ivan Russo, Emil Durukan, Maria Giovanna Asmundo, Arturo Lo Giudice, Serena Salzano, Sebastiano Cimino, Antonio Rescifina, Mikkel Fode, Ali Saber Abdelhameed, Rosario Caltabiano, Giuseppe Broggi

**Affiliations:** 1Urology Section, Department of Surgery, University of Catania, 95123 Catania, Italy; mariagiovannaasmundo@gmail.com (M.G.A.); arturologiudice@gmail.com (A.L.G.); scimino@unict.it (S.C.); 2Department of Urology, Copenhagen University Hospital, Herlev Hospital, 2730 Copenhagen, Denmark; emildurukan@gmail.com (E.D.); mikkelfode@gmail.com (M.F.); 3Department of Medical and Surgical Sciences and Advanced Technologies “G. F. Ingrassia”, Anatomic Pathology, University of Catania, 95123 Catania, Italy; sere.salzano@gmail.com (S.S.); rosario.caltabiano@unict.it (R.C.); giuseppe.broggi@gmail.com (G.B.); 4Department of Drug Sciences, University of Catania, 95125 Catania, Italy; arescifina@unict.it; 5Department of Pharmaceutical Chemistry, College of Pharmacy, King Saud University, P.O. Box 2457, Riyadh 11451, Saudi Arabia; asaber@ksu.edu.sa

**Keywords:** prostate cancer, CYP7B1, biochemical recurrence, prognosis, Warburg

## Abstract

We aimed to analyze the association between CYP7B1 and prostate cancer, along with its association with proteins involved in cancer and metabolic processes. A retrospective analysis was performed on 390 patients with prostate cancer (PC) or benign prostatic hyperplasia (BPH). We investigated the interactions between CYP7B1 expression and proteins associated with PC and metabolic processes, followed by an analysis of the risk of biochemical recurrence based on CYP7B1 expression. Of the 139 patients with elevated CYP7B1 expression, 92.8% had prostate cancer. Overall, no increased risk of biochemical recurrence was associated with CYP7B1 expression. However, in a non-diabetic subgroup analysis, higher CYP7B1 expression indicated a higher risk of biochemical recurrence, with an HR of 1.78 (CI: 1.0–3.2, *p* = 0.05). PC is associated with elevated CYP7B1 expression. In a subgroup analysis of non-diabetic patients, elevated CYP7B1 expression was associated with an increased risk of biochemical recurrence, suggesting increased cancer aggressiveness.

## 1. Introduction

Prostate cancer (PC) is the second most commonly diagnosed cancer in men. In 2020, about 1.4 million cases of prostate cancer were reported globally [1]. Androgens, such as dihydrotestosterone (DHT) acting through the androgen receptor (AR), play a critical role in the development and growth of prostate cancer by promoting cell proliferation in the prostate [2]. In addition to the production of DHT, the conversion of DHT into weaker androgen metabolites known as 5α-androstane-3β,17β-diol (3βAdiol) and 5α-androstane-3α,17β-diol (3αAdiol) regulates androgen levels within the prostate [3]. This metabolic process is catalyzed by an enzyme called CYP7B1, which belongs to the cytochrome P450 family. CYP7B1 is primarily expressed in the liver and is involved in the metabolism of steroid hormones [4], playing a critical role in regulating the cell cycle, as well as the proliferative, invasive, and migratory activity of cancer cells [5,6].

CYP7B1 is involved in metabolism of a number of steroids reported to influence estrogen and androgen signaling [7,8].

It has been observed that an increased expression of CYP7B1 is associated with more aggressive forms of prostate cancer, indicating a potential link between this enzyme and the progression of the disease [9]. An alteration was observed by Olsson et al. in the ratio of estrogen receptor beta (ERβ) to CYP7B1 mRNA in the tumor regions, suggesting a potential disruption in estrogen signaling within prostate cancer cells [9]. Consequently, the combination of low levels of 3βAdiol (a weaker androgen metabolite) and increased CYP7B1 expression, along with a decreased expression of ERβ, is unfavorable and likely to play a significant role in the initial stages of prostate cancer development [9]. Furthermore, Lutz et al. discovered a link between CYP7B1 expression and tumor content in diabetic men, suggesting a connection to changes in insulin/IGF-1 receptors. The study also revealed a positive correlation between CYP7B1 expression and androgen signaling activity, measured by the KLK3 gene-encoding prostate-specific antigen (PSA), as well as an association with cell proliferation indicated by Ki-67 gene expression [10]. In this study, we aimed to analyze the association between CYP7B1 and prostate cancer, along with its association with proteins involved in cancer and metabolic processes.

## 2. Results

A total of 286 patients were diagnosed with PC and underwent radical prostatectomy, while 104 patients were diagnosed with benign prostatic hyperplasia (BPH) at the final histology. Table 1 displays the clinical characteristics of both groups. The number of patients with diabetes in PC and BPH patients was 44 (15.38%) and 28 (26.92%), respectively.

Appendix A shows clinical and pathological data in the sub-cohort of patients without diabetes.

### 2.1. Increased Expression of CYP7B1 in Patients with PC

In the immunohistochemical analysis, we observed a higher expression of CYP7B1 in malignant prostate tissue compared to benign prostate tissue. Among the 139 patients with elevated CYP7B1 expression, 92.8% were diagnosed with prostate cancer. Furthermore, we found a significant increase in the expression of CYP7B1 in cases with a pathological stage of T4 among patients with PC (17.8%; *p* = 0.04) (Table 2).

### 2.2. The Association between CYP7B1 and Other Proteins Involved in Cancer and Metabolic Processes

We observed a significant association between increased CYP7B1 expression and higher levels of PSMA, IR-α, IR-β, SRSF-1, CPT1-a, SREBP1, FAS, and ACC-1 (Table 2).

In our logistic regression model, we identified a significantly higher probability of CYP7B1-positive expression for IR-α and IR-β, with odds ratios (OR) of 5.73 (confidence interval (CI): 2.77–11.84, *p* < 0.01) and 6.61 (CI: 2.19–19.96, *p* < 0.01), respectively. Furthermore, we found a significant positive association between increased CYP7B1 expression and SRSF-1 and FAS, with ORs of 2.04 (CI: 1.27–3.29, *p* < 0.01) and 2.15 (CI: 1.28–3.62, *p* < 0.01), respectively. Additionally, there was a positive association between higher CYP7B1 expression and elevated PSMA (OR 1.66, CI: 1.04–2.66, *p* = 0.03) and ACC-1 (OR 1.83, CI: 1.14–2.93, *p* = 0.01) (Table 3).

Table 4 and Table 5 show the univariate logistic regression between immunohistochemistry results and clinical and pathological variables in PC patients. In particular, the expression of CYP7B1 was associated with a positive expression of PSMA (OR: 3.15, CI: 2.04–4.85; *p* < 0.05), CPT1a (OR: 1.98, CI: 1.15–3.39, <0.05), and FAS (OR: 3.35, CI: 2.14–5.25, *p* < 0.05), and a reduced expression of SREPB (OR: 0.59, CI: 0.36–0.99, *p* < 0.05).

In the multivariate logistic regression analysis (adjusted for age, PSA, CPT1a, ATPLy, SREBP, FAS), CYP7B1 expression was not associated with ISUP ≥ 4 (OR: 1.69; CI: 0.73–3.90; 0.22) and pathological stage T3/4 (OR: 1.12; CI: 0.64–1.96; *p* = 0.70).

We performed multiple multivariable models in order to predict ISUP ≥ 4 and pathological stage ≥ pT3-4 and we did not find significant independent variables after adjusting for age and PSA.

### 2.3. Association between CYP7B1 Expression and the Risk of Biochemical Recurrence in Patients with PC

There was no elevated risk of biochemical recurrence linked to CYP7B1 expression in patients overall (HR: 1.39; *p* = 0.21). However, in a subgroup analysis of patients without diabetes (n = 58), CYP7B1-positive expression was positively associated with BCR (age-adjusted HR: 1.775; CI: 1.001–3.169, *p* = 0.05).

Biochemical recurrence (BCR)-free survival at 5 years was 79% (CI 69–86%) and 59% (CI 46–70%) in CYP7b1-negative and CYP7b1-positive patients, respectively (Figure 1).

Overall, the competing risk analysis did not demonstrate any statistically significant association between CYP7B1 (subhazard ratio (SHR): 1.18; *p* = 0.55) and age (SHR: 0.99; *p* = 0.64), but did demonstrate a positive association with pathological stage (pT3/4 vs. pT2) (SHR: 1.80; 95%CI 1.00–3.29; *p* < 0.05) and ISUP Gleason (≥4 vs. <4) (SHR: 2.44; 95%CI 1.17–5.08; *p* < 0.05). Figure 2 shows the cumulative incidence function for BCR according to CYP7B1 expression.

Finally, in the non-diabetic population with ISUP <4 (n = 206), the age-adjusted competing risk analysis of CYP7B1 was 1.94 (95%CI 0.98–3.86; *p* = 0.058). Figure 3 shows the cumulative incidence function for BCR according to CYP7B1 expression.

## 3. Discussion

In the present study, we aimed to explore the relationships between the expression of certain genes (CYP7B1, SRSF-1, FAS, PSMA, ACC-1, CPT1a, and SREBP) and oncological factors in prostate cancer patients with and without diabetes. The results indicate significant associations between the expression of these genes and specific markers, as well as clinical and pathological variables. For instance, we found that the expression of CYP7B1, a gene involved in cholesterol metabolism, was significantly associated with the expression of insulin receptor subtypes IR-α and IR-β. The odds ratios (ORs) for positive CYP7B1 expression in relation to IR-α and IR-β were 5.73 and 6.61, respectively. Additionally, increased CYP7B1 expression was positively associated with the expression of SRSF-1 and FAS, with ORs of 2.04 and 2.15, respectively. Furthermore, there were positive associations between higher CYP7B1 expression and elevated levels of PSMA and ACC-1, with ORs of 1.66 and 1.83, respectively. Notably, the expression of CYP7B1 was associated with a positive expression of PSMA, CPT1a, and FAS, while it was inversely associated with SREBP expression. These associations were quantified using odds ratios, which provide a measure of the strength and direction of the relationship between gene expression and other variables.

In recent years, there has been growing interest in understanding the influence of modifiable risk factors on the development and progression of PC, particularly regarding the metabolic pathways that are involved in PC aggressiveness [11], and understanding the potential histological factors associated with oncological outcomes [12]. In our immunohistochemical analysis of malignant and benign prostate tissue, we revealed an increased expression of CYP7B1 in men with prostate cancer and an association between an increased risk of biochemical recurrence in a subgroup analysis of men without diabetes. It is important to highlight that the association we found was slightly close to the level of significance and it merits careful attention, given that in the competing risk analysis, we did not find such a relationship.

Furthermore, CYP7B1 was associated with the altered expression or activity of insulin receptors (IR-α and IR-β), along with a potential involvement in pathways related to mRNA splicing (SRSF-1); fatty acid synthesis (FAS and ACC-1); and ATPLy, CPT1a, SREBP, FAS, and PSMA expression in prostate cancer.

Studies conducted with CYP7B1-/- and ERβ-/- knockout mice revealed that CYP7B1 is able to abolish ERβ-mediated anti-proliferative effects via the metabolism of the ERβ ligand 5a- androstane-3b,17b-diol, which is reported to counteract AR-mediated proliferation [13]. Tang et al. concluded that DHT is not only a precursor of 5a-androstane-3b,17b-diol, but may also control 5a-andro- stane-3b,17b-diol metabolism via effects on the CYP7B1 gene [14]. These results suggest that androgens may be able to regulate estrogen levels within the prostate, a previously unknown relationship between androgens and estrogens within the prostate [14].

We have previously shown that on PC patients, positive SRSF-1 expression was associated with AR, Ki-67, IR-α, and microvascular density, indicating a potential role of SRSF-1 in driving tumor aggressiveness [15]. Furthermore, a Kaplan–Meier analysis revealed that patients with positive SRSF-1 expression had worse 5- and 9-year biochemical recurrence rates compared to those with negative SRSF-1 expression. Moreover, when analyzing patients with both PC and diabetes, positive SRSF-1 expression was associated with increased levels of Ki-67 and MVD, suggesting a link between SRSF-1 expression, cell proliferation, and angiogenesis in the context of diabetes-associated PC. The study findings suggest that PC exhibits heterogeneous protein expression patterns, with SRSF-1 expression playing a significant role in tumor aggressiveness and prognosis. Additionally, this study highlights a potential association between altered insulin signaling, protein expression patterns, and tumor progression, particularly in patients with diabetes. These results provide valuable insights for future research aimed at understanding the molecular mechanisms underlying PC progression and identifying potential therapeutic targets [15,16].

Weihua et al. [17] showed that 5a-androstane-3b,17b-diol (3βAdiol) is an endogenous ER ligand. Cytochrome P450 (CYP) 7B1 and 3b-hydroxysteroid dehydrogenase (3β-HSD) are two enzymes that determine the levels of 3βAdiol available in the ER and, thus, may influence the control of prostate proliferation. CYP7B knockout mice show lower prostate proliferation rates compared to WT mice [13]. Polymorphisms in the 3β-HSD gene are associated with prostate cancer risk [18]. Jakobsson et al. [19] screened the human CYP7B1 gene for possible polymorphisms. Only one single polymorphism was detected, a C–G change in the promoter –104 base pair from the transcription start site. The allele frequency was investigated in Swedish men and was compared to a Korean population, and the authors found that that the frequency of the G-allele was 4.04% in Swedes, but only 0.33% among Koreans. This polymorphism is associated with phenotypic differences in expression systems and vastly different allele frequencies in the two ethnic populations, resulting in large differences in prostate cancer incidence.

Lutz et al. [10] showed a positive correlation of CYP7B1 expression with tumor content only in the samples in men with diabetes, while men without diabetes showed the opposite direction. CYP7B1 expression positively correlated with the activity of androgen signaling and Ki-67 gene expression. Therefore, the downregulation of synthases or overexpression of degradative enzymes may cause a decrease in these protective steroids and a shift from estrogen to androgen signaling [20].

Interestingly, Kakiyama et al. reported data showing that an inability to upregulate CYP7B1, in the setting of insulin resistance, results in the accumulation of toxic intracellular cholesterol metabolites that promote inflammation and hepatocyte injury [21]. The authors provided strong evidence that the insulin resistance-mediated dysregulation of CYP7B1 establishes a cellular mechanism for the accumulation of “toxic” intracellular cholesterol metabolites [21]. These data showed that insulin resistance alters this pathway by leading to a chronically lower expression of CYP7B1 [21]. As a result, the chronic accumulation of oxysterols appears not only to directly provide substrate for toxic metabolites, but also to alter the regulation of pathways such as mitochondrial cholesterol uptake and FFA synthesis [22].

Although controversial, these data may justify the finding of our study that revealed a higher risk of biochemical recurrence among individuals with higher levels of CYP7B1 expression in a subgroup analysis of patients without diabetes (HR = 1.78). In conclusion, a deeper understanding of the relationship between lipid metabolism and PC progression is of great importance and should be explored in the near future to improve survival in PC patients. Finally, we would like to mention some limitations. First, we were unable to examine the association between body weight composition and lipid expression. Second, we have not examined the link between metabolism and possible genomic alterations. Third, the short follow-up period in our cohort was virtually inadequate to assess overall survival. On the other hand, our study is one of the few studies that reported tissue changes in CYP7B1 metabolism in PC and its association with progression and prognosis. The selection process for patients undergoing radical prostatectomy or surgery for benign prostatic hyperplasia (BPH) inherently introduces bias due to the varied nature of patient presentations and treatment options. This bias arises from the fact that not all individuals diagnosed with prostate cancer or BPH will undergo surgery, as some may opt for alternative treatments like radiation therapy or active surveillance, or remain undiagnosed altogether. Consequently, the decision to proceed with surgery is influenced by numerous factors such as disease severity, patient preferences, comorbidities, and physician recommendations, resulting in a non-random selection that can skew outcomes and the interpretations of our results.

## 4. Materials and Methods

### 4.1. Study Design

We conducted a retrospective analysis of data from 390 patients who underwent either radical prostatectomy for prostate cancer or transurethral resection of the prostate (TURP) for benign prostatic hyperplasia (BPH) at the Department of Urology, University of Catania, between 2010 and 2020. This study was conducted according to The Code of Ethics of the World Medical Association (Declaration of Helsinki) and was reviewed and approved by the local Ethics Committee. Informed consent was obtained from the study participants. The protocol was approved by the Ethics Committee of the Policlinic Hospital of Catania (#131/2015).

### 4.2. Immunohistochemistry

Hematoxylin and eosin staining were performed on all specimens to identify representative cores for the construction of the tissue microarray. The Galileo TMA CK3500 (Integrated System Engineering, Milan, Italy) was utilized for the tissue microarray (TMA) construction, as previously described [11,15,16]. This semi-automatic and computer-assisted tissue microarrayer comes with dedicated software that guides the user through the entire process, from designing the tissue microarray to generating the final report. The instrument is equipped with an X-Y-Z automated stage, allowing the precise placement of selected tissue cores into premade holes in the recipient TMA block. This feature not only significantly reduces the time required for array construction, but also ensures precise alignment.

Immunohistochemical slides were assessed by three pathologists (G.B., E.P., and R.C.) who were blinded to the patients’ clinical data, following the previously described methodology [11]. Immunohistochemical analyses were carried out using the established methodology [11]. We analyzed the expression of the protein CYP7B1 using immunohistochemical analysis in both malignant and benign prostatic tissue and investigated the risk of prostate cancer biochemical recurrence in relation to CYP7B1 expression. Additionally, we explored the association and interactions between increased CYP7B1 expression and proteins involved in cancer and metabolic processes, including Ki-67, androgen receptor (AR), prostate-specific membrane antigen (PSMA), insulin receptor α subunit (IRα), insulin receptor beta subunit (IRβ), insulin-like growth factor 1 receptor (IGF-1R), serine/arginine-rich splicing factor 1 (SRSF-1), carnitine palmitoyltransferase 1-α (CPT1-a), stearoyl-CoA desaturase-1 (SCD-1), sterol regulatory element-binding protein 1 (SREBP1), fatty acid synthase (FAS), and acetyl-CoA carboxylase-1 (ACC-1).

### 4.3. Statistical Analysis

Continuous variables are presented as median and interquartile range (IQR) and were compared by Student’s independent t-test or the Mann–Whitney U test based on their normal or not-normal distribution, respectively (the normality of the variables’ distribution was tested by the Kolmogorov–Smirnov test). Categorical variables were analyzed using a chi-squared test. Univariate logistic regression was employed to examine independent variables associated with IHC scores. Logistic regression is a statistical method used for modeling the relationship between one or more independent variables (predictors) and a categorical dependent variable with two or more categories. In logistic regression, the outcome variable is modeled using the logistic function, which ensures that the predicted values fall between 0 and 1, representing probabilities. The odds ratio (OR) is often used to interpret the impact of predictor variables on the outcome. It is calculated as the exponentiation of the coefficient. An odds ratio greater than 1 suggests that the odds of the event increase as the predictor variable increases. An odds ratio less than 1 suggests that the odds of the event decrease as the predictor variable increases.

Prostate cancer (PC) was categorized as low, intermediate, or high based on the European Association of Urology guidelines [23]. Kaplan–Meier curves were generated to assess the correlation between IHC scores and the probability of biochemical recurrence-free survival. Statistical significance among the curves was evaluated using a log-rank test. Biochemical recurrence (BCR) was defined as the occurrence of a PSA value exceeding 0.2 ng/mL during the follow-up period. A competing risk analysis has been performed to correctly estimate the marginal probability of an event in the presence of competing events (i.e., age, diabetes, CYP7B1 expression).

Differences between groups were considered significant at a *p*-value less than 0.05. Data analysis was conducted using StataCorp. 2021 (Stata Statistical Software: Release 3024).

## 5. Conclusions

A deeper understanding of the relationship between lipid metabolism and PC progression is of great importance and should be explored in the near future to improve survival in PC patients. CYP7B1 may be considered a crucial marker for PC aggressiveness and further studies may be focused on better understanding the mechanisms of its intraprostatic modulation. It may serve as a potential biomarker for predicting disease progression and the response to androgen-targeted therapies. Additionally, targeting CYP7B1 or its downstream pathways could represent a novel therapeutic strategy for managing aggressive forms of prostate cancer. Further studies with a bigger sample and longer follow-up are needed in order to better estimate the impact of metabolism on CYP7B1 expression and its role in PC progression and mortality.

The association between CYP7B1 and prostate cancer aggressiveness is multifaceted and involves the modulation of androgen metabolism, intratumoral androgen levels, and the interaction with steroid hormone pathways. Further research is needed to elucidate the precise mechanisms underlying this association and to explore its therapeutic implications.

## Figures and Tables

**Figure 1 ijms-25-04762-f001:**
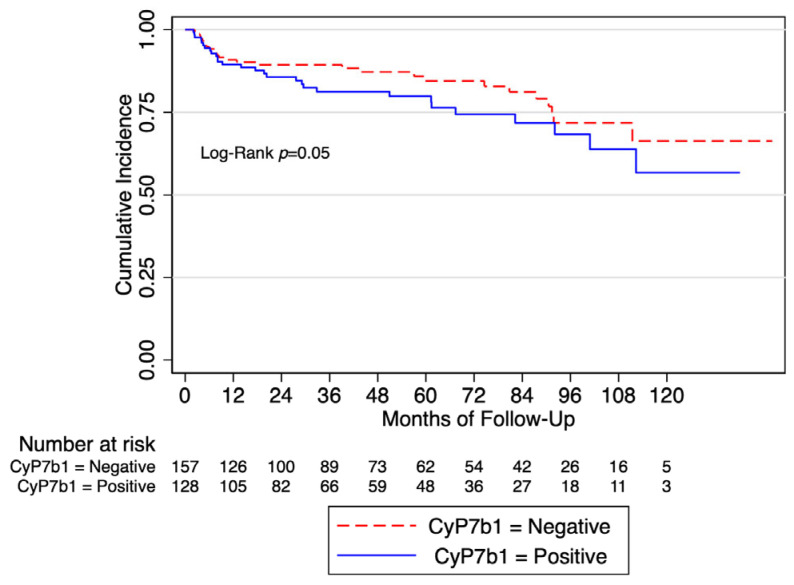
Biochemical recurrence-free survival in the whole cohort according to the expression of CYP7b1.

**Figure 2 ijms-25-04762-f002:**
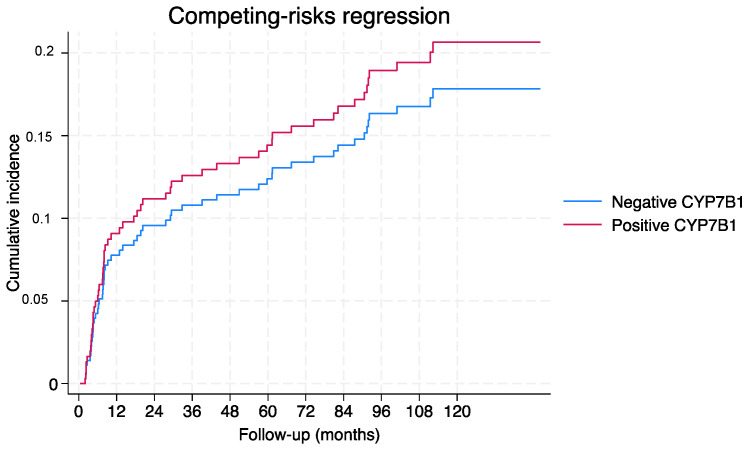
Overall cumulative incidence function for BCR according to CYP7B1 expression.

**Figure 3 ijms-25-04762-f003:**
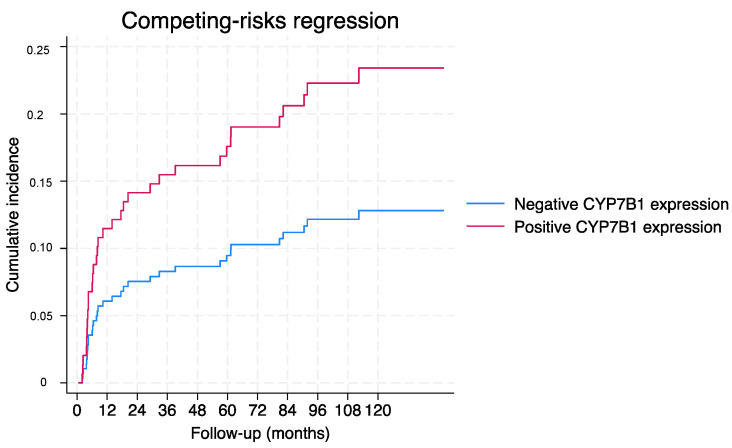
Cumulative incidence function for BCR according to CYP7B1 expression in the non-diabetic population with ISUP < 4.

**Table 1 ijms-25-04762-t001:** Baseline characteristics of the final cohort.

	PC (n = 286)	BPH (n = 104)	*p*-Value
Age (years median)	68 (63–72)	78 (72–81)	<0.01
Diabetes, n (%)	44 (15.38)	28 (26.92)	0.01
Fasting blood glucose (mg/dL), median (Q1–Q3)	97 (88–107)	95 (84–116)	0.71
Total cholesterol (mg/dL), median (Q1–Q3)	191.5 (168–221.5)	175 (150–199)	<0.01
Triglycerides (mg/dL), median (Q1–Q3)	97 (68–130)	117 (69–164)	<0.01
PSA (ng/mL), median (Q1–Q3)	7.8 (5.7–11.6)	2.05 (0.87–4.32)	<0.01
BCR, n (%)	56 (19.6)	-	

PC = prostate cancer; BPH = benign prostatic hyperplasia; PSA = prostate-specific antigen; BCR = biochemical recurrence.

**Table 2 ijms-25-04762-t002:** CYP7B1 expression according to IHC score.

	CYP7B1		*p*-Value
	Negative (n = 251)	Positive (n = 139)	
Age (years), median (Q1–Q3)	71 (65–76)	68 (63–71)	<0.01
PSA (ng/mL), median (Q1–Q3)	6.1 (3.53–9.6)	7.2 (5.4–11)	<0.01
Fasting glucose (mg/dL), median (Q1–Q3)	96 (87–109)	97 (88–111)	0.9
Total cholesterol (mg/dL), median (Q1–Q3)	181.5 (156–210)	190.5 (165.5- 218)	0.1
Triglycerides (mg/dL), median (Q1–Q3)	100 (68–137)	103 (72–146)	0.9
Diabetes, n (%)	59 (23.5)	33 (23.74)	0.95
Group, n (%)			<0.01
BPH	94 (37.45)	10 (7.19)	
PC	157 (62.55)	129 (92.81)	
ISUP Gleason score in PC patients, n (%)			0.25
1	49 (31.21)	33 (25.58)	
2	63 (40.13)	47 (36.43)	
3	34 (21.66)	32 (24.81)	
4	4 (2.55)	10 (7.75)	
5	7 (4.46)	7 (5.43)	
Pathological stage in PC patients, n (%)			0.04
T2	111 (70.70)	86 (66.67)	
T3	33 (21.02)	20 (15.50)	
T4	13 (8.28)	23 (17.83)	
Classification risk of PC, n (%)			0.2
Low risk	56 (35.67)	44 (34.11)	
Intermediate risk	74 (47.13)	52 (40.31)	
High risk	27 (17.20)	33 (25.58)	
Ki-67-positive score, n (%)	31 (12.35)	22 (15.83)	0.3
AR-positive score, n (%)	110 (43.82)	72 (51.80)	0.1
PSMA-positive score, n (%)	71 (28.29)	77 (55.40)	<0.01
IR-α-positive score, n (%)	132 (52.59)	127 (91.37)	<0.01
IR-β-positive score, n (%)	4 (1.59)	19 (13.67)	<0.01
IGF-1R-positive score, n (%)	36 (14.34)	28 (20.14)	0.1
ATPLy-positive score, n (%)	92 (36.65)	95 (68.35)	<0.01
SRSF-1-positive score, n (%)	99 (39.44)	89 (64.03)	<0.01
CPT1-a-positive score, n (%)	33 (13.15)	32 (23.02)	0.01
SCD-1-positive score, n (%)	39 (15.54)	26 (18.71)	0.4
SREBP1-positive score, n (%)	70 (27.89)	26 (18.71)	0.04
FAS-positive score, n (%)	111 (44.22)	101 (72.66)	<0.01
ACC-1-positive score, n (%)	69 (27.49)	75 (53.96)	<0.01

IHC = immunohistochemistry; BPH = benign prostatic hyperplasia; PC = prostate cancer; Q1–Q3 = interquartile range; CYP7B1 = cytochrome 7B1; AR = androgenic receptor; PSA = prostate-specific antigen; IR = insulin receptor; IGF-1R = insulin-like growth factor-1 receptor; PSMA = prostate-specific membrane antigen; SRSF-1 = aerine/arginine-rich splicing factor 1; FAS = fatty acid synthase; CPT-1a = carnitine palmitoyltransferase 1a; SCD-1 = stearoyl-CoA desaturase-1; SREBP-1 = sterol regulatory element-binding protein-1; ACC-1 = acetyl-CoA carboxylase-1.

**Table 3 ijms-25-04762-t003:** Univariate logistic regression of association between CYP7B1 and clinical and pathological variables in the whole cohort.

	Positive Expression of CYP7B1	*p*-Value
Age, OR (CI)	1.00 (0.97–1.04)	0.8
PSA, OR (CI)	1.01 (0.99–1.03)	0.3
Fasting glucose, OR (CI)	1.00 (0.99–1.02)	0.6
Total cholesterol, OR (CI)	1.00 (0.99–1.01)	0.6
Triglycerides, OR (CI)	1.00 (0.99–1.00)	0.7
Diabetes, OR (CI)	0.91 (0.47–1.74)	0.8
Ki-67-positive score, OR (CI)	0.91 (0.49–1.67)	0.8
AR-positive score, OR (CI)	1.36 (0.85–2.17)	0.2
PSMA-positive score, OR (CI)	1.66 (1.04–2.66)	0.03
IR-α-positive score, OR (CI)	5.73 (2.77–11.84)	<0.01
IR-β-positive score, OR (CI)	6.61 (2.19–19.96)	<0.01
IGF-1R-positive score, OR (CI)	0.93 (0.53–1.63)	0.8
ATPLy-positive score, OR (CI)	3.73 (2.40–5.79)	<0.01
SRSF-1-positive score, OR (CI)	2.04 (1.27–3.29)	<0.01
CPT1-a-positive score, OR (CI)	1.33 (0.74–2.41)	0.3
SCD-1-positive score, OR (CI)	1.22 (0.67–2.21)	0.5
SREBP1-positive score, OR (CI)	0.61 (0.35–1.06)	0.08
FAS-positive score, OR (CI)	2.15 (1.28–3.62)	<0.01
ACC-1-positive score, OR (CI)	1.83 (1.14–2.93)	0.01

OR = odds ratio; CI = confidence interval; CYP7B1 = cytochrome 7B1; AR = androgenic receptor; PSA = prostate-specific antigen; IR = insulin receptor; IGF-1R = insulin-like growth factor-1 receptor; PSMA = prostate-specific membrane antigen; SRSF-1 = serine/arginine-rich splicing factor 1; FAS = fatty acid synthase; CPT-1a = carnitine palmitoyltransferase 1a; SCD-1 = stearoyl-CoA desaturase-1; SREBP-1 = sterol regulatory element-binding protein-1; ACC-1 = acetyl-CoA carboxylase-1.

**Table 4 ijms-25-04762-t004:** Univariate logistic regression between clinical and oncological outcomes and IHC results for ATPLy, CPT1a, SCD, and SREBP.

	ATPLy + vs. − (OR 95% CI)	CPT-1a, + vs. − (OR 95% CI)	SCD + vs. − (OR 95% CI)	SREBP + vs. − (OR 95% CI)
**PSA, continuous**	1.01 (0.98–1.03)	1.00 (0.97–1.02)	0.98 (0.95–1.01)	0.96 (0.92–1.00)
**Fasting blood glucose, continuous**	0.99 (0.98–1.01)	1.00 (0.99–1.02)	0.99 (0.98–1.01)	1.00 (0.99–1.01)
**Total cholesterol, continuous**	0.99 (0.98–1.00)	0.99 (0.98–1.01)	0.99 (0.98–1.00)	0.99 (0.98–1.00)
**Triglycerides, continuous**	1.00 (0.99–1.01)	1.00 (0.99–1.01)	0.99 (0.98–1.00)	0.99 (0.98–1.00)
**Diabetes, yes vs. no**	1.11 (0.58–2.16)	0.82 (0.49–2.43)	1.58 (0.74–3.39)	0.51 (0.22–1.21)
**Pathological stage, pT3/4 vs. pT2**	1.27 (0.76–2.12)	1.08 (0.79–2.04)	0.94 (0.49–1.80)	1.04 (0.60–1.85)
**ISUP Gleason, ≥4 vs. <4**	1.82 (0.77–4.30)	0.75 (0.44–3.02)	1.53 (0.61–3.82)	0.79 (0.30–2.04)
**AR, + vs. −**	1.71 (1.06–2.77) ^†^	2.27 (1.24–4.16) ^†^	2.87 (1.53–5.39) ^†^	2.16 (1.25–3.73) ^†^
**PSMA, + vs. −**	1.12 (0.70–1.80)	0.97 (0.54–1.75)	1.16 (0.64–2.12)	0.94 (0.55–1.61)
**Ki-67, + vs. −**	1.33 (0.71–2.50)	1.37 (0.66–2.84)	2.16 (1.07–4.32) ^†^	1.02 (0.51–2.04)
**IR-α, + vs. −**	2.56 (1.43–4.56) ^†^	2.55 (1.03–6.27) ^†^	1.20 (0.56–2.56)	1.93 (0.92–4.05)
**IR-β, + vs. −**	1.08 (0.45–2.59)	2.45 (1.01–6.11) ^†^	1.24 (0.44–3.51)	1.33 (0.52–3.38)
**IGF-1R, + vs. −**	1.30 (0.73–2.32)	0.96 (0.47–1.95)	1.01 (0.49–2.07)	0.35 (0.16–0.78) ^†^
**ATPLy + vs. −**	-	1.26 (0.69–2.32)	1.43 (0.76–2.68)	1.41 (0.81–2.47)
**CPT-1a, + vs. −**	1.26 (0.69–2.32)	-	2.15 (1.08–4.24) ^†^	2.95 (1.58–5.49) ^†^
**SCD + vs. −**	1.43 (0.76–2.68)	2.15 (1.08–4.24) ^†^	-	2.87 (1.53–5.39) ^†^
**SREBP -1+ vs. −**	1.41 (0.81–2.47)	2.95 (1.57–5.48) ^†^	2.87 (1.53–5.39) ^†^	-
**FAS + vs. −**	4.84 (2.84–8.25) ^†^	2.16 (1.05–4.41) ^†^	3.17 (1.42–7.04) ^†^	1.74 (0.94–3.21)
**CYP7B1 + vs. −**	3.73 (2.40–5.79) ^†^	1.98 (1.15–3.39) ^†^	1.25 (0.72–2.16)	0.59 (0.36–0.99) ^†^

OR = odds ratio; PSA = Prostate specific antigen; CI = confidence interval; CYP7B1 = cytochrome 7B1; AR = androgenic receptor; PSA = prostate-specific antigen; IR = insulin receptor; IGF-1R = insulin-like growth factor-1 receptor; ATPLy = ATP citrate lyase; PSMA = prostate-specific membrane antigen; FAS = fatty acid synthase; CPT-1a = carnitine palmitoyltransferase 1aSCD-1 = stearoyl-CoA desaturase-1; SREBP-1 = sterol regulatory element-binding protein-1; ^†^ *p* < 0.05.

**Table 5 ijms-25-04762-t005:** Univariate logistic regression between clinical and oncological outcomes and IHC results for FAS, ACC-1, and CYP7B1.

	FAS + vs. − (OR 95% CI)	ACC-1 + vs. − (OR 95% CI)	CYP7B1 + vs. − (OR 95% CI)
**PSA, continuous**	1.00 (0.98–1.03)	0.99 (0.97–1.01)	1.20 (0.93–1.54)
**Fasting blood glucose, continuous**	0.99 (0.98–1.00)	0.99 (0.98–1.01)	0.99 (0.98–1.01)
**Total cholesterol, continuous**	0.99 (0.99–1.00)	0.99 (0.98–1.00)	1.01 (0.99–1.01)
**Triglycerides, continuous**	0.99 (0.99.1.00)	0.99 (0.98–1.00)	1.00 (0.99–1.04)
**Diabetes, yes vs. no**	0.50 (0.26–0.97)	1.60 (0.83–3.06)	0.82 (0.47–1.41)
**Pathological stage, pT3/4 vs. pT2**	0.71 (0.42–1.20)	1.30 (0.93–1.80)	1.20 (0.72–1.98)
**ISUP Gleason, ≥4 vs. <4**	1.47 (0.60–3.60)	1.21 (0.98–1.52)	2.01 (0.91–4.47)
**AR, + vs. −**	2.19 (1.30–3.69) ^†^	3.65 (2.22–5.93) ^†^	1.37 (0.91–2.09)
**PSMA, + vs. −**	1.64 (1.00–2.71) ^†^	1.80 (1.13–2.89) ^†^	3.15 (2.04–4.85) ^†^
**Ki-67, + vs. −**	1.67 (0.83–3.38)	1.11 (0.60–2.03)	1.33 (0.74–2.41)
**IR-α, + vs. −**	3.31 (1.84–5.95) ^†^	9.99 (4.35–22.93) ^†^	9.54 (5.02–18.12) ^†^
**IR-β, + vs. −**	1.77 (0.63–4.95)	1.85 (0.77–4.43)	9.78 (3.25–29.37) ^†^
**IGF-1R, + vs. −**	0.80 (0.44–1.44)	1.18 (0.67–2.05)	1.50 (0.87–2.59)
**ATPLy + vs. −**	4.84 (2.84–8.25) ^†^	4.97 (2.95–8.39) ^†^	3.73 (2.40–5.79) ^†^
**CPT-1a, + vs. −**	2.16 (1.05–4.41) ^†^	2.12 (1.16–3.87) ^†^	1.98 (1.15–3.39) ^†^
**SCD + vs. −**	3.17 (1.42–7.04) ^†^	2.63 (1.40–4.91) ^†^	1.25 (0.72–2.16)
**SREBP-1 + vs. −**	1.74 (0.94–3.21)	2.53 (1.45–4.40) ^†^	0.59 (0.36–0.99) ^†^
**FAS + vs. −**	-	11.29 (5.76–22.14) ^†^	3.35 (2.14–5.25) ^†^
**CYP7B1 + vs. −**	3.35 (2.14–5.25) ^†^	3.09 (2.00–4.77) ^†^	-

OR = odds ratio; PSA = prostate specific antigen; CI = confidence interval; CYP7B1 = cytochrome 7B1; AR = androgenic receptor; PSA = prostate-specific antigen; IR = insulin receptor; IGF-1R = insulin-like growth factor-1 receptor; PSMA = prostate-specific membrane antigen; FAS = fatty acid synthase; CPT-1a = carnitine palmitoyltransferase 1a; SCD-1 = stearoyl-CoA desaturase-1; SREBP-1 = sterol regulatory element-binding protein-1; ^†^ *p* < 0.05.

## Data Availability

Data can be requested from the corresponding author.

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
