# Peer review of "CYP7B1 as a Biomarker for Prostate Cancer Risk and Progression: Metabolic and Oncogenic Signatures (Diagnostic Immunohistochemistry Analysis by Tissue Microarray in Prostate Cancer Patients—Diamond Study)"

_ijms, 2024, doi:10.3390/ijms25094762_

Round 1
Reviewer 1 Report
Comments and Suggestions for Authors
Ref: ijms-2960719
Title: The Association Between CYP7B1 and Diabetes in Prostate Cancer patients (DIAgnostic Immunohistochemistry analysis By tissue Microarray in prOstate caNcer patients – DIAMOND STUDY)
Journal: International Journal of Molecular Sciences
Reviewer’s comments
The topic addressed by this paper (an analysis of the association between prostate cancer and CYP7B1 enzyme) is noteworthy and interesting; the study appears well planned; the methods and statistical analysis are clearly described and correctly applied (but the calculation of sample size and statistical power were not shown); the results are consequent and presented with sufficient clearness (there are only some aspects that can be improved); the discussion is well developed (though some considerations could be added).
So, in my opinion the paper could be suitable for publication. Anyway, I have some points to signal, hoping they could be useful to further improve the quality of the paper. In the following I report some comments on the manuscript.
I have no conflicts of interest to declare.
Specific comments:
1. The placement of Materials and Methods section after the Discussion is atypical: I suggest to move it between the Introduction and the Results to improve the readability and comprehensibility of the paper.
2. The acronym PCa must be uniform throughout the paper (sometimes it is used PCA, other times PC).
3. The same applies to the acronym CYP7B1, which several times indicated as CYP7b1 and once CYP7B (line 188).
4. Has a calculation of sample size and statistical power been performed? If yes, why this information was not reported in Materials and Methods section?
5. The term IQR should be replaced by Q1-Q3 (in the text and in tables 1 and 2), as the authors did not report the interquartile range (which is a single value), but the first and third quartiles.
6. The acronym BCR was not defined at its first appearance (but anticipating the Materials and Methods section as suggested this problem would be solved).
7. Line 75: “…we found a significant increase in the expression of CYP7B1…”; the reference to the significance test used and its p-value are missing.
8. Tables 2, 3, 4 and 5. please revise carefully the correspondence between the acronyms used and their definitions in the footnotes, as some of them are not defined, some defined but not used, some are written in a different way.
9. In tables 4 and 5, the symbol † (presumably indicating statistically significant ORs) should be defined.
10. Lines 121-124: the size of the subgroup of non-diabetic patients is not mentioned, though it can obtained from Table 1 (286-44+104-28=318); by the way, it would be better to write “in the subgroup analysis”. Furthermore, the evidence of significance is so low that it coincides with the significance level of 0.05, as similarly indicated by the lower limit of the HR confidence interval, which is equal to 1; such data deserves to be addressed in the Discussion section, also given that the "age adjusted competing risk analysis" investigated it better and excluded evidence of significance (p-value=0.058, lower limit of the confidence interval=0.98).
11. Line 136: “…in the non-diabetic population with ISUP <4…”; here too, the size of the subgroup must be declared, given that it is a subgroup of non-diabetic patients; this information cannot be obtained from the tables due to the possible intersections between subgroups.
12. Line 137: “Figure 4 shows…” must be changed into “Figure 3 shows…”.
13. Lines 185-221: as these paragraphs illustrate the state of the art on CYP7B1, they should be moved to Introduction section, and only referred to in Discussion.
Author Response
We would like to thank the reviewer for its time dedicated to revise our paper and it prestigious comments.
We have revised the paper according to the typos you mentioned. We updated the discussion regarding the point you have raised.
As concerning the methods, this section is put after the discussion based on instruction of the journal.
Reviewer 2 Report
Comments and Suggestions for Authors
Thank you very much for the opportunity to review this paper. This study presents an important topic and, for this reason, its results should be carefully presented. However, many important questions still need to be resolved in order to consider this manuscript to publication:
1. The manuscript is titled “The Association Between CYP7B1 and Diabetes in Prostate Cancer Patients”. In line 21, the authors state their aim to analyze the association between CYP7B1 and prostate cancer, along with its association with proteins involved in cancer and metabolic processes.
The objectives presented in the title and in the Abstract are different. Please clarify in the manuscript whether the objective is to study the association between CYP7B1 and diabetes in prostate cancer patients or the association between CYP7B1 and prostate cancer.
2. Why was not a control population of healthy male patients considered in this study? Would it not be necessary to consider a control population of healthy men if the objective was to evaluate the association between CYP7B1 and prostate cancer? In the discussion section (line 234), the authors themselves emphasize that the use of the BPH group could introduce bias (consequently, distort the results).
3. Throughout the manuscript, the authors used multiple acronyms to refer to the same definition. For instance, they used the acronyms PC, PCA and PCa to refer to “prostate cancer”. Please, standardize this.
4. Table 1. Please, define the meaning of BCR (this definition was presented only in line 294). Furthermore, by definition, the interquartile range (IQR) is calculated as the difference between Q3 and Q1. Therefore, the IQR should be presented as a single value rather than an interval. Please fix this.
5. In table 1, it is suggested that the authors perform a hypothesis test in order to compare the values between the considered groups.
6. Based on data from Section 2.1, it was found that 45% (129/286) of prostate cancer patients and 9.6% (10/104) of BPH patients had higher expression of CYP7B1. It is suggested that the authors compare this value with the literature and with a healthy population.
7. The section 2.1 says “Furthermore, we found a significant increase in the expression of CYP7B1 in cases with a pathological stage of T4 among patients with PCA”. Authors are advised to include a table that presents the expression of CYP7B1 in relation to different stages and to conduct an appropriate hypothesis test to support this statement.
8. Section 2.2: The values presented in this table appear to be incorrect. For example, based on the table, the presented values for BPH and PC are represented as n and %. How is it possible for the value of n to be non-integer? Assuming that the value 37.5 is correct, 37.5/251 = 14.9%, not 3.05%. The Ki-67, AR, and PSMA variables exhibit the same behavior. Please, clarify/fix the presented results.
9. Please, define the meaning of ATPL in Table 4. It is suggested that the authors provide a more detailed discussion of the results obtained in Tables 4 and 5. Please, discuss the differences found and the important results that do not show a difference.
10. The authors are advised to sort the acronyms presented below the tables to improve reader visualization.
11. Please, add an additional decimal place to the results presented in line 123: 1.78 (CI: 1.0-3.2, p=0.05). The statement “in a subgroup analysis of patients without diabetes, we identified a higher risk of biochemical recurrence among individuals with higher levels of CYP7B1 expression, indicated by a HR (age adjusted) of 1.78” is unclear to me.
12. As the authors presented a non-diabetic patients analysis, it would be important to include a descriptive analysis for this group. What are the age and stage distributions, as well as the ISUP Gleason distribution, for non-diabetic patients with elevated and non-elevated levels of CYP7B1?
13. Please, increase the size of Figure 1.
14. Please, include bibliographic references to support the statement in lines 167-170.
15. Lines 277-278 says “Continuous variables are presented with their median and interquartile range (IQR) and were assessed for non-normal distribution using the Mann-Whitney U test”. The Mann-Whitney U test is not suitable for assessing normality of the distribution. This test is used to compare the medians between two independent groups, when the data do not have a normal distribution. Please, clarify.
16. Lines 279-280 says “Univariate logistic regression was employed to examine independent variables associated with IHC scores”. On table 3, were the results based on various single regressions?
Multiple regression is preferred over single regression in certain situations because it enables for the examination of the relationship between a dependent variable and multiple independent variables simultaneously. This statement provides a more comprehensive understanding of the factors that influence the dependent variable.
In the manuscript analysis, did the authors perform many single logistic regressions instead of a multiple logistic regression? In this case, it would be more appropriate to use multiple logistic regression.
Comments on the Quality of English Language
Extensive English language editing is required.
Author Response
We would like to thank the reviewer for his time in revising this paper and his prestigious comments. Please find below our replies:
- The manuscript is titled “The Association Between CYP7B1 and Diabetes in Prostate Cancer Patients”. In line 21, the authors state their aim to analyze the association between CYP7B1 and prostate cancer, along with its association with proteins involved in cancer and metabolic processes.
The objectives presented in the title and in the Abstract are different. Please clarify in the manuscript whether the objective is to study the association between CYP7B1 and diabetes in prostate cancer patients or the association between CYP7B1 and prostate cancer.
We updated the title according to this suggestion.
- Why was not a control population of healthy male patients considered in this study? Would it not be necessary to consider a control population of healthy men if the objective was to evaluate the association between CYP7B1 and prostate cancer? In the discussion section (line 234), the authors themselves emphasize that the use of the BPH group could introduce bias (consequently, distort the results).
We would like to thank you for this comment. However, collecting samples from the prostate of healthy men is impossible for many reasons, including the ethical one. This is the reason why we collected from BPH patients.
- Throughout the manuscript, the authors used multiple acronyms to refer to the same definition. For instance, they used the acronyms PC, PCA and PCa to refer to “prostate cancer”. Please, standardize this.
We have corrected the abbreitations.
- Table 1. Please, define the meaning of BCR (this definition was presented only in line 294). Furthermore, by definition, the interquartile range (IQR) is calculated as the difference between Q3 and Q1. Therefore, the IQR should be presented as a single value rather than an interval. Please fix this.
Thank you so much. We have updated the definition in the table 1 and we updated the Q1-Q3 definition.
- In table 1, it is suggested that the authors perform a hypothesis test in order to compare the values between the considered groups.
In table 1 we added the p-value.
- Based on data from Section 2.1, it was found that 45% (129/286) of prostate cancer patients and 9.6% (10/104) of BPH patients had higher expression of CYP7B1. It is suggested that the authors compare this value with the literature and with a healthy population.
We would like the thank you for this comment. CYP7B1 is an enzyme involved in the cholesterol metabolims in the healthy population with no specific other actions.
- The section 2.1 says “Furthermore, we found a significant increase in the expression of CYP7B1 in cases with a pathological stage of T4 among patients with PCA”. Authors are advised to include a table that presents the expression of CYP7B1 in relation to different stages and to conduct an appropriate hypothesis test to support this statement.
We have already included this data in table 2.
- Section 2.2: The values presented in this table appear to be incorrect. For example, based on the table, the presented values for BPH and PC are represented as n and %. How is it possible for the value of n to be non-integer? Assuming that the value 37.5 is correct, 37.5/251 = 14.9%, not 3.05%. The Ki-67, AR, and PSMA variables exhibit the same behavior. Please, clarify/fix the presented results.
We apologise for these mistakes. We corrected the table.
- Please, define the meaning of ATPL in Table 4. It is suggested that the authors provide a more detailed discussion of the results obtained in Tables 4 and 5. Please, discuss the differences found and the important results that do not show a difference.
We updated the table accordingly. As concerning the discussion, we highlighted the findings related to CYP7B1 expression and other markers.
- The authors are advised to sort the acronyms presented below the tables to improve reader visualization.
We have updated the table with abbreviations.
- Please, add an additional decimal place to the results presented in line 123: 1.78 (CI: 1.00-3.17, p=0.05). The statement “in a subgroup analysis of patients without diabetes, we identified a higher risk of biochemical recurrence among individuals with higher levels of CYP7B1 expression, indicated by a HR (age adjusted) of 1.78” is unclear to me.
We updated that part accordingly to your suggestions.
- As the authors presented a non-diabetic patients analysis, it would be important to include a descriptive analysis for this group. What are the age and stage distributions, as well as the ISUP Gleason distribution, for non-diabetic patients with elevated and non-elevated levels of CYP7B1?
We have added a suppl. table 1.
- Please, increase the size of Figure 1.
Done
- Please, include bibliographic references to support the statement in lines 167-170.
Done.
- Lines 277-278 says “Continuous variables are presented with their median and interquartile range (IQR) and were assessed for non-normal distribution using the Mann-Whitney U test”. The Mann-Whitney U test is not suitable for assessing normality of the distribution. This test is used to compare the medians between two independent groups, when the data do not have a normal distribution. Please, clarify.
We have updated the section for a better comphrension.
- Lines 279-280 says “Univariate logistic regression was employed to examine independent variables associated with IHC scores”. On table 3, were the results based on various single regressions?
Multiple regression is preferred over single regression in certain situations because it enables for the examination of the relationship between a dependent variable and multiple independent variables simultaneously. This statement provides a more comprehensive understanding of the factors that influence the dependent variable.
In the manuscript analysis, did the authors perform many single logistic regressions instead of a multiple logistic regression? In this case, it would be more appropriate to use multiple logistic regression.
Yes data on table 3 were univariate. As concerning the multivariate, since the CYP7B1 expression was not associated with pathological and other clinical data we did not perform the multivariate. Obviously you are right about this comment and we agree with your observation. Unfortunately, data were not strong enough to justify a multivariable model.
Round 2
Reviewer 2 Report
Comments and Suggestions for Authors
I would like to thank the authors for answering my questions and making the suggested changes in the manuscript. However, I consider that some important points still need to be evaluated in detail:
Comment 01, first review:
Authors’ answer: We updated the title according to this suggestion.
Reply: Authors are still encouraged to revise the title of the manuscript.
Comment 07, first review:
Authors’ answer: We have already included this data in table 2.
Reply: It seems to me that this information is not presented in table 2. There are 23 individuals who are positive for CYP7B1. It is not mentioned if they are in the PC or BPH group. Then, the information “we found a significant increase in the expression of CYP7B1 in cases with a pathological stage of T4 among patients with PC” cannot be obtained from table 2.
Comment 11, first review:
Authors’ answer: We updated that part accordingly to your suggestions.
Reply: I believe that these values were not updated as suggested.
Comment 16, first review:
Authors’ answer: Yes data on table 3 were univariate. As concerning the multivariate, since the CYP7B1 expression was not associated with pathological and other clinical data we did not perform the multivariate. Obviously you are right about this comment and we agree with your observation. Unfortunately, data were not strong enough to justify a multivariable model.
Reply: What does it mean “data were not strong enough to justify a multiple model”? What are the data criteria considered for this classification? Have the authors tried to make a model with multiple variables? If so, in the final model construction, did the authors exclude the non-significant variables one by one (greater p-value) until only significant variables remain in the model? It is possible that, if the independent variables are related, the exclusion of one of them returns another as significant. The use of various univariate models does not seem to be appropriate to me.
Comments on the Quality of English LanguageModerate English language editing is required.
Author Response
Comment 01, first review:
Authors’ answer: We updated the title according to this suggestion.
Reply: Authors are still encouraged to revise the title of the manuscript.
We provided to change the title accordingly.
Comment 07, first review:
Authors’ answer: We have already included this data in table 2.
Reply: It seems to me that this information is not presented in table 2. There are 23 individuals who are positive for CYP7B1. It is not mentioned if they are in the PC or BPH group. Then, the information “we found a significant increase in the expression of CYP7B1 in cases with a pathological stage of T4 among patients with PC” cannot be obtained from table 2.
We apologize for the misunderstanding. Pathological stage is related to an oncological diagnosis that only belong to patients affected by PC. For this reason, we changed the raw title for a better comprehension.
Comment 11, first review:
Authors’ answer: We updated that part accordingly to your suggestions.
Reply: I believe that these values were not updated as suggested.
We changed to CYP7B1 positive expression was positively associated with BCR (age adjusted HR: 1.775; CI: 1.001-3.169, p=0.05).
Comment 16, first review:
Authors’ answer: Yes data on table 3 were univariate. As concerning the multivariate, since the CYP7B1 expression was not associated with pathological and other clinical data we did not perform the multivariate. Obviously you are right about this comment and we agree with your observation. Unfortunately, data were not strong enough to justify a multivariable model.
Reply: What does it mean “data were not strong enough to justify a multiple model”? What are the data criteria considered for this classification? Have the authors tried to make a model with multiple variables? If so, in the final model construction, did the authors exclude the non-significant variables one by one (greater p-value) until only significant variables remain in the model? It is possible that, if the independent variables are related, the exclusion of one of them returns another as significant. The use of various univariate models does not seem to be appropriate to me.
We performed multiple multivariable models in order to predict ISUP ≥4 and pathological stage ≥ pT3-4 and we did not find significant independent variables after adjusting for age and PSA. We updated results by adding these sentence.
The lack of significant results may be related to the low number of events in our cohort.
English has been revised.
Round 3
Reviewer 2 Report
Comments and Suggestions for Authors
I would like to thank the authors for answering my questions and making the suggested changes in the manuscript.
-
The manuscript title is better now.
-
In this phase of the assessment process, I read the recent publications of the authors regarding the DIAMOND study. In reference [11], I located the results (OR 95% CI) of Table 4 (with the exception of the values displayed in the final line, "CYP7B1 + vs. -") and of Table 5 (with the exception of the values displayed in the final column, "CYP7B1 + vs. -").
As these results have already been presented and discussed in the previous article (reference [11]), and the discussion regarding these tables in this manuscript (lines 89-107) was not based on these repeated results, please exclude them.
[11] Russo GI, Asmundo MG, Lo Giudice A, Trefiletti G, Cimino S, Ferro M, et al. Is There a Role of Warburg Effect 360 in Prostate Cancer Aggressiveness? Analysis of Expression of Enzymes of Lipidic Metabolism by 361 Immunohistochemistry in Prostate Cancer Patients (DIAMOND Study). Cancers (Basel) 2023;15:948. 362 https://doi.org/10.3390/cancers15030948.
Please accept my apologies for not identifying this occurrence at an earlier stage.
Author Response
We would like to thank the reviewer for his time dedicating in revising this paper.
However, results were not previously presented in other papers. The study has an huge amount of data. Data on publication on Cancer were without CYP7b1 expression that is the main core of the current paper. Each protein expression requires months of work and interpretation, statistical analysis and comparison between other variables.
Indeed, if we look at all previous work, there is no mention of CYP7b1 that has been for the first time investigated here and also in the current literature by the way.
We hope to have fulfilled all comments.
Round 4
Reviewer 2 Report
Comments and Suggestions for Authors
I would like to thank the authors for answering my questions. I consider that the article is ready for publication.